# Co-Cultures of *Lactobacillus acidophilus* and *Bacillus subtilis* Enhance Mucosal Barrier by Modulating Gut Microbiota-Derived Short-Chain Fatty Acids

**DOI:** 10.3390/nu14214475

**Published:** 2022-10-25

**Authors:** Zhengjun Xie, Meng Li, Mengqi Qian, Zhiren Yang, Xinyan Han

**Affiliations:** 1Hainan Institute, Zhejiang University, Yongyou Industry Park, Yazhou Bay Sci-Tech City, Sanya 572000, China; 2College of Animal Science, Zhejiang University, Hangzhou 310058, China; 3Twins Group Corporation, Nanchang 330096, China

**Keywords:** probiotic, intestinal barrier, mucosal immunity, gut microbiota, short-chain fatty acids

## Abstract

Weaning stress induces intestinal barrier dysfunction and immune dysregulation in mammals. Various interventions based on the modulation of intestinal microbiota have been proposed. Our study aims to explore the effects of co-cultures from *Lactobacillus acidophilus* and *Bacillus subtilis* (FAM^®^) on intestinal mucosal barrier from the perspective of metabolic function of gut microbiota. A total of 180 piglets were allocated to three groups, i.e., a control group (C, basal diet), a FAM group (F, basal diet supplemented with 0.1% FAM), and an antibiotic group (A, basal diet supplemented with antibiotic mixtures). Here, we showed FAM supplementation significantly increased body weight and reduced diarrhea incidence, accompanied by attenuated mucosal damage, increased levels of tight junction proteins, serum diamine oxidase (DAO) and antimicrobial peptides. In addition, 16S rRNA sequencing and metabolomic analysis revealed an increase in relative abundance of Clostridiales, Ruminococcaceae, Firmicutes and Muribaculaceae and a significant increase in the total short-chain fatty acids (SCFAs) and butyric acid in FAM-treated piglets. FAM also increased CD4^+^ T cells and SIgA^+^ cells in intestinal mucosa and SIgA production in colon contents. Furthermore, FAM upregulated the expression of IL-22, short-chain fatty acid receptors GPR43 and GPR41, aryl hydrocarbon receptor (AhR), and hypoxia-inducible factor 1α (HIF-1α). FAM shows great application prospect in gut health and provides a reference for infant weaning.

## 1. Introduction

Intestinal barrier functions as a selective construction to prevent environmental antigens invasion, which is critical for immune resistance and host survival [1]. The development of intestinal barrier occurs rapidly after birth and is characterized by decreased gut permeability [2]. Epithelial permeability barrier is mainly regulated by the tight junctions which consist of intracellular and apical intercellular membrane proteins, including zonula occludens (ZO), claudins and occluding [3,4,5], which regulate epithelial leakiness by selectively modulating ion and pore size of the intestinal epithelium [5]. Specialized epithelial cell types such as goblet cells and Paneth cells could also support intestinal barrier function by providing protective mucous layer and secreting antimicrobial peptides [6,7]. Weaning stress induces intestinal barrier dysfunctions, including defects of intestinal epithelial junction, decreased thickness of mucosal layer, and defective production of antimicrobial peptides [8].

Intestinal microbiota composition and activity co-evolve with the host from birth and is influenced by nutrition and lifestyle [9]. A million years of coevolution have led to a symbiotic relationship between microbiota and host in which gut microbiota likely mediate host physiology and metabolism [9,10,11]. Specifically, microbiota-induced cell signaling lead to changes in the mucosa barrier function, immune response and metabolic pathway, thereby affecting host physiology and pathophysiology [12,13]. Most of human diseases, such as diarrhea, Crohn’s disease (CD), irritable bowel syndrome (IBS) and obesity, are associated with dysbiosis of gut microbiota and loss of microbial diversity [14,15]. A wide use of antibiotics and other factors may reduce the number of bacterial predators, leading to a decrease in intestinal microbial diversity [16,17]. Considering the complex interplay between gut microbiota and host health, interventions based on the modulation of gut microbiota have been considered one of the most potential. In recent years, increasing studies have demonstrated that probiotic microorganisms confer a health benefit in humans and animals, including interaction with resident microbiota [18,19], modulation of immune function [20], production of antimicrobial compounds [21] and organic acids [22,23], and improving gut barrier integrity [24].

Pigs have many similarities with humans in the intestinal composition and function, making the pig an ideal animal model [25]. With regard to fermenting dietary fibrous composition, pigs and humans both use the colon rather than the cecum as the main site [25]. Furthermore, porcine and humans share 96% similarity in gastro-intestinal microbiota functional pathways [26]. Here, we use a piglet model to study the in vivo mechanism of FAM^®^ (co-cultures of *Lactobacillus acidophilus* and *Bacillus subtilis*) on intestinal barrier function from the perspective of gut microbiota. The effect of FAM on gut barrier was associated with alteration in gut microbiota and butyrate metabolic processes. Our study may give new insights in helping to understand the modulation of probiotic ferment on gut health and their potential mechanisms underlying in animals or humans.

## 2. Materials and Methods

### 2.1. Animals and Experimental Design

The animal experiment was approved by the Animal Care and Use Committee of Zhejiang University (SYXK 2012- 0178) and all experimental procedures conformed to the institutional guideline for animal study. A total of 180 piglets (Duroc ×Landrace × Yorkshire hybrid) weaned at 28 days old with similar initial body weight were randomly allocated to three groups, i.e., a control group (C, basal diet), a FAM group (F, basal diet supplemented with 0.1% FAM), and an antibiotic group (A, basal diet supplemented with antibiotic mixtures). Each group has four replicates (i.e., pens) and fifteen piglets per replicate. All piglets were housed in pens and had free access to feed and water. The three groups were independently housed in separated region and all environmental conditions of the three regions were kept consistently. The basal diet was designed to meet the nutrient requirements of the National Research Council (NRC, 2016) for weaned piglets. FAM^®^ (provided by Zhejiang Kangwandechuan Technology Co., Ltd., Shaoxing, China) is co-fermented by *Lactobacillus acidophilus* (≥1 × 106 CFU/g) and *Bacillus subtilis* (≥1 × 106 CFU/g). Antibiotic mixtures contained 50 mg/kg quinocetone, 55 mg/kg kitasamycin, 75 mg/kg chlortetracycline, and 200 mg/kg oxytetracycline. After a 7 days adaptation period, piglets were fed their respective diet for a 30-day experimental period. The body weight, feed consumption, and diarrhea incidence of each piglet were observed and recorded. The schematic diagram for the animals and experimental design was seen as below (Figure 1).

### 2.2. Sample Collection

At the end of the feeding trial, eight piglets per group (two pig from each replicate with average body weight) were randomly selected and humanely killed after 12 h fasting. Blood samples were collected and centrifuged at 3000× *g* at 4 °C for 15 min to obtain the serum. Mucosa samples from ileum and colon were collected and snap-frozen in liquid nitrogen and then stored at −80 °C. The intestinal tissues (1 × 1 cm^2^) from jejunum and ileum were collected and fixed in 4% paraformaldehyde for histomorphology and immunohistochemistry analysis. The jejunal tissues (0.5 × 0.5 cm^2^) were collected and fixed in 2.5% glutaraldehyde fixative for electron microscopy. The colonic contents (middle part) were collected in 1.5 mL sterile centrifuge tubes and snap-frozen in liquid nitrogen and stored at −80 °C for microbiome and metabolite analyses.

### 2.3. Intestinal Histomorphology

After dehydration, jejunal and ileal samples were imbedded in paraffin wax and cut into 5 μm sections using a microtome. The sections were further stained with hematoxylin and eosin (H&E) and Periodic Acid-Schiff for morphological analysis and images were acquired using an Olympus BX 51 microscope (Olympus Corporation, Tokyo, Japan). The number of goblet cells (PAS) was measured with computer-assisted microscopy (Micrometrics TM; Nikon ECLIPSE E200, Tokyo, Japan). Transmission electron microscopy and scanning electron microscopy visualization were conducted according to the previous study [27].

### 2.4. Gene Expression Determined by Real-Time Quantitative PCR (qRT-PCR)

Relative mRNA expressions of Porcine beta defensin-2 (PBD-2), Porcine beta defensin-3 (PBD-3) and Regenerating islet-derived IIIγ (RegⅢ γ) in the ileum mucosa were determined by qRT-PCR using the designed primers (shown in Appendix A. Quantitative analysis was performed with the Power SYBR Green PCR Master Mix (Applied Biosystems) on the CFX384 real-time fluorescence quantitative PCR system. GAPDH was used as a housekeeping gene and the relative expression of target gene was analyzed based on the 2−ΔΔCt method.

### 2.5. Western Blot (WB)

Approximately 100 mg of tissues was dissolved in T-PER Tissue Protein Extraction Reagent (including Protease Inhibitor Cocktail) for protein extraction. The concentrations of total protein were measured with BCA protein assay kit. Relative protein expressions were determined by WB according to the previous study. The primary antibodies used in this study were as follows: MUC2 (Novus NB120-11197, Littleton, CO, USA), ZO-1 (Thermo Fisher 40-2200, Waltham, MA, USA), Claudin1 (Abcam ab129119, Cambridge, UK), Occludin (Abcam ab222691, Cambridge, UK), GPR43 (Proteintech 19952-1-AP, Wuhan, China), GPR41 (Abcam ab103718, Cambridge, UK), HIF-1α (Thermo Fisher MA1-516, Waltham, MA, USA), AhR (Novus Biologicals NB100-2289, Littleton, CO, USA), IL-22 (Affinity DF8343, Changzhou, China) and GADPH (Abcam ab181602, Cambridge, UK). Relative levels of target protein were normalized against GAPDH.

### 2.6. Enzyme-Linked Immunosorbent Assay (ELISA)

Protein levels of diamine oxidase (DAO), endotoxin (ET) and D-lactate in serum and SIgA in ileum content were detected by pig ELISA kits (Nanjing Jiancheng Bioengineering Institute, Nanjing, China) following the manufacturer’s recommendations. The concentrations of total protein were measured by BCA protein assay kit (Beyotime Institute of Biotechnology, Shanghai, China) to normalize the target protein concentration.

### 2.7. Immunohistochemistry

The density of SIgA+ cells and CD4+ T cells were analyzed by immunohistochemistry according to the previous study. Briefly, paraffin-embedded ileum sections were deparaffinized and rehydrated. After unmasking antigens, the sections were stained with SIgA (Pig IgA Antibody, 1:1000; Bethyl Laboratories, Montgomery, TX, USA) and CD4 (Rabbit CD4 Antibody, 1:500; Servicebio, GB11064) primary antibodies overnight at 4 °C and then treated by secondary antibody with streptavidin-horseradish peroxidase. Finally, signals were detected by diaminobenzidine.

### 2.8. Colonic Content Microbiome Analysis

The microbiome genomic DNA of colon contents was extracted using QIAamp Stool DNA mini kits (Qiagen, New York, NY, USA) following the manufacturer’s instructions, and then variable region 4 (V4) of 16s rRNA gene was amplified by polymerase chain reaction (PCR). The PCR products were mixed and subsequently purified with Qiagen Gel Extraction Kits (Qiagen). Sequencing library was generated using TruSeq DNA PCR-free Sample Preparation Kits (Illumina) according to the manufacturer’s recommendations and sequenced on the Illumina HiSeq2500 platform to obtain 250 bp paired-end reads, which further merged with FLASH (Version 1.2.7, http://ccb.jhu.edu/software/FLASH/) to generate raw tags. Effective tags were generated for further analysis after data filtration and chimera removal. Sequences with > 97% similarity were classified to the same operational taxonomic unit (OTU) by the Uparse software (Version 7.0.1001, http://drive5.com/uparse/), and the taxonomic information for representative sequence was annotated using GreenGene Database. Multiple sequence alignment was conducted on the MUSCLE software (Version 3.8.31, http://www.drive5.com/muscle/) to determine the phylogenetic relationship of different OTUs and the dominant species of different groups. The selected predictive functions were conducted using Spearman’s correlation analysis (IBM SPSS Inc., Chicago, IL, USA).

### 2.9. Untargeted Metabolomic Analysis

Metabolomic analysis of colonic contents was conducted by multiple mass spectrometry (MS) platforms, including gas chromatography mass spectrometry/time-of-flight (GC-MS/TOF) and ultrahigh-performance liquid chromatography/mass spectrometry (UHPLC/MS). GC-MS/TOF analysis was performed on the Agilent 7890A gas chromatograph system coupled with the Pegasus HT TOF MS (Leco) while UHPLC/MS analysis was conducted on the Agilent 1290 UHPLC system coupled to TripleTOF 6600 system (Q-TOF, AB Sciex, Concord, ON, Canada). Further multivariate statistical analysis was performed on the SIMCA software (Version 14.1, MKS Data Analytics Solutions, Concord, ON, Canada). Group differences and group separation variables were analyzed using Orthogonal projections to latent structures-discriminate analysis (OPLS-DA). The predictive ability parameter Q2 and goodness-of-fit parameter R2Y were obtained for estimating the model quality after seven-fold cross validation. The metabolite set enrichment analysis and pathway analysis were carried out to generate the related Kyoto Encyclopedia of Genes and Genomes (KEGG) pathway of each differential metabolite and biomarker metabolic pathways, separately, on the web-based tool MetaboAnalyst (http://www.meta-boanalyst.ca, accessed on 17 March 2021).

### 2.10. Targeted Metabonomic Analysis

Quantification of short-chain fatty acid (Acetic acid, propionic acid, butyric acid, isobutyric acid, valeric acid, isovaleric acid and caproic acid) in the colonic contents was analyzed using the following procedures. Briefly, MS analysis of targeted metabolite was performed using the Agilent 1290 Infinity series UHPLC System, equipped with a Waters ACQUITY UPLC BEH amide column or a Waters ACQUITY UPLC BEH C18 column. The standard curve was obtained by subjecting the standard solution with UPLC-parallel reaction monitoring (PRM)-MS/MS analysis. Finally, the extracted ion chromatographs (EICs) of targeted analyte from the samples and the standard solution were obtained and the metabolite concentration of colonic content in each sample was calculated (nmol/g).

### 2.11. Statistical Analysis

Data in this article are expressed as means ± standard error of the means (SEM). Statistical differences between two groups were analyzed by Mann–Whitney U-test or by unpaired Student’s *t*-test, and data among three groups were evaluated by one-way ANOVA, followed by Tukey’s multiple comparisons (SPSS 23.0). Probability values *p* < 0.05 were considered statistically significant and 0.05 < *p* < 0.10 were considered a trend.

Microbial richness was measured based on richness indices (Chao1, observed species, ACE) and microbial diversity was accessed by diversity indices (Simpson and Shannon). Anosim analysis was performed using the Bray–Curtis methods. β-diversity was determined based on the principal coordinate analysis (PCoA) and non-metric multidimensional scaling (NMDS). Relative abundance of the microbial flora was expressed as median percentages. Linear discriminant analysis effect size (LEfSe) analysis was accessed on online LEfSe tool (http://huttenhower.sph.harvard.edu/galaxy, accessed on 21 September 2021).

## 3. Results

### 3.1. FAM Supplementation Promoted Intestinal Barrier Function

As shown in Table 1, FAM and antibiotics significantly increased body weight and decreased diarrhea incidence of weaned piglets. In addition, increased number of microvilli and mitochondria, and restored epithelial junctions were observed in FAM and antibiotic group, indicating FAM or antibiotics improved intestinal morphology (Figure 1A). Furthermore, intestinal barrier function in weaned piglets was measured. As shown in Figure 1B,D, compared to control and antibiotic-treated piglets, FAM-treated piglets had higher expression levels of tight junction proteins (ZO-1, and Occludin) and MUC2 protein as well as higher number of goblet cells. Relative mRNA expression of antimicrobial peptides (AMPs) was analyzed by qPCR, involving porcine beta defensin-2 (PBD-2), porcine beta defensin-3 (PBD-3) and regenerating islet-derived IIIγ (RegIIIγ) (Figure 1E). The results showed that FAM or antibiotics significantly upregulated the expression of PBD-2, PBD-3 and RegIIIγ, and PBD-2 expression in FAM group was much higher than that of in antibiotics. To further clarify intestinal epithelial functions were improved following FAM supplementation, the levels of serum endotoxin, DAO and D-lactate were detected. As shown in Figure 1F, the level of DAO was significantly decreased following FAM treatment (Figure 1G), demonstrating FAM improved intestinal barrier function and intestinal permeability.

### 3.2. FAM Regulated Microbial Diversity and Structure in the Colon

To detect the direct influence of FAM on the richness and diversity of the gut microbiome, α-diversity analysis was performed. As shown in Figure 2A, observed species, Chao1, ACE and PD whole tree in FAM group were significantly increased compared with those in antibiotic group, suggesting FAM could increase the richness and diversity of intestinal microbiome. To examine the composition alteration of gut microbiota, β-diversity analysis was conducted. The cluster tree of UPGMA Cluster Analysis indicated a significant dissimilarity in the composition of intestinal microbiota among three groups at the phylum level (Appendix A). The difference of the colonic microbiome among three groups was also confirmed by separately clustered gut microbiota shown in PCoA and NMDS analysis (Figure 2B).

To explore the specific changes in the structure of gut microbiota, relative abundance of microbial community was analyzed at three different taxonomic levels (Appendix A). At the phylum level, Firmicutes (75.16%) and Bacteroidetes (22.67%) represented the two most dominant microbiota. The relative abundance of Proteobacteria in FAM group (0.32%) and antibiotic group (0.35%) was lower than that in control group (0.52%), indicating FAM and antibiotics could reduce harmful bacteria in colon. At the family level, *Ruminococcaceae*, *Prevotellaceae*, *Lachnospiraceae*, and unidentified_clostridiales were the predominant taxon. Notably, FAM group displayed increased *Muribaculaceae* abundance, indicating FAM might promote intestinal functions for degrading complex carbohydrates. At the genus level, *Terrisporobacter*, *Bacteroides*, *Blautia*, *Faecalibacterium*, and *Lactobacillus* are predominant taxon. LEfSe analysis further revealed FAM markedly increased the relative abundance of 18 bacterial biomarkers including *Clostridiales*, *Ruminococcaceae*, *Firmicutes* and *Muribaculaceae* and decreased relative abundance of 22 microbial biomarkers involving swine *Streptococcus*, *Bacteroides*, and *Megamonas* (Figure 2C).

### 3.3. Functional Metagenomics Prediction and Spearman’s Correlation Analysis of Gut Microbiota

To explore whether FAM-induced microbial changes modulate the metabolic function of gut microbiota, functional metagenomics prediction of gut microbiota was conducted based on 16S rRNA gene sequencing (Figure 3A–C). Our results showed 18 pathways at the third level of KEGG pathway were significantly altered following FAM supplementation, including significantly increased proportions of ‘butanoate metabolism’, ‘pyruvate metabolism’, and ‘propanoate metabolism’ and significantly decreased proportions of ‘peptidases’, ‘sphingolipid metabolism’, and ‘chaperones and folding catalysts’. Moreover, compared to antibiotics-treated piglets, the proportions of ‘butanoate metabolism’, ‘pyruvate metabolism’, ‘ABC transporters’, ‘propanoate metabolism’, and ‘tryptophan metabolism’ were markedly increased and proportions of ‘lipopolysaccharide biosynthesis’ and ‘lipopolysaccharide biosynthesis proteins’ were decreased in FAM-treated piglets. To further analyze the correlation between significantly different genera and specific predictive functions, spearman’s correlation analysis was performed. As shown in Figure 3D, *phyla Firmicutes* and *genera Clostridium* were positively correlated with ‘butanoate metabolism’, ‘pyruvate metabolism’, ‘tryptophan metabolism’, and ‘fatty acid metabolism’ and negatively correlated with ‘lipopolysaccharide biosynthesis’. Meanwhile, *phyla Bacteroidetes*, Tenericutes, and Helotiales were negatively correlated with ‘lipopolysaccharide biosynthesis’ and positively correlated with ‘butanoate metabolism’ and ‘pyruvate metabolism’.

### 3.4. FAM Induced Functional Changes of Intestinal Metabolome in the Colon Contents

The changes of microbiota metabolic function usually result in the changes in metabolites. To exam functional changes of intestinal metabolome in piglets, the colon contents were analyzed by GC-MS. OPLS-DA score plots showed that intestinal metabolome clustered separately among three groups and volcano plots further revealed the significantly different metabolites between groups (Figure 4A,B). As demonstrated in Figure 4C, the levels of palmitoleic acid, lithocholic acid and squalene in FAM group was much higher than those in control and antibiotic group. Moreover, we observed significantly higher concentration of amines (cadaverine, maleimide and o-phosphorylethanolamine) in antibiotic group compared to those in FAM group (Figure 4D). To further identify biomarker KEGG pathways, metabolite set enrichment analysis was performed. The results showed that amino acid metabolism, aminoacyl-tRNA biosynthesis and short chain fatty acid metabolism were significantly affected following FAM treatment (Figure 4E), which was inconsistent with the function prediction of the gut microbiota (Figure 3A,B) that butanoate metabolism was significantly increased in FAM group. Therefore, we subsequently explored the modulatory role of FAM in butanoate metabolism of gut microbiota.

### 3.5. FAM Promoted Butyric Acid Production and Enhanced Mucosal Immune Function

To further confirm the alteration in butanoate metabolism detected by the untargeted metabolomic analysis, LC-MS/MS-based targeted metabolomic approach was used to quantify the precise concentration of short-chain fatty acids (SCFAs), including acetic acid, propionic acid, iso-butyric acid, butyric acid, isovaleric acid and valeric acid. As shown in Figure 5A, the levels of butyric acid and total SCFAs in the FAM group were much higher than other groups. Compared to the control and FAM group, the antibiotic group had lower content of total SCFAs, acetic acid and butyric acid. SCFAs are the major metabolic products of microbiota from dietary fiber and have been recognized as important mediators in regulating mucosal immunity and intestinal homeostasis [28]. Therefore, we detected mucosal immune functions and inflammatory responses. The results showed that FAM increased the integrated optical density (IOD) of CD4+ T cells and SIgA+ cells in intestinal mucosa as well as SIgA production in colon contents, suggesting FAM enhanced mucosal immunity (Figure 5B–D). Compared to the antibiotic group, the FAM group had lower expression level of proinflammatory gene NF-κB and TNF-α and higher level of anti-inflammatory gene IL-22 (Figure 5E), which was in accordance with previous studies that intestinal microbiota-derived SCFAs could regulate CD4+ T cell and gut immunity [28]. Compared to the control group, antibiotic group had higher expression of proinflammatory gene TNF-α and lower level of anti-inflammatory gene IL-22, indicating the adverse effects of antibiotics on piglets.

### 3.6. FAM Upregulated IL-22 Production and Enhanced GPR41 and GPR43 Activation

From the above results, we concluded that FAM enhanced intestinal barrier function and upregulated intestinal microbiota derived SCFAs production. To further demonstrate how FAM regulated intestinal barrier, we evaluated the expression short-chain fatty acid receptors including GPR43, GPR41and GPR109A. The results demonstrated that the relative expression of GPR43 and GPR41 in the FAM group was much higher than the control and antibiotic group (Figure 6A). Based on the changes of GPR43, GPR41 and IL-22, we further determined their relative protein expression. Consistently, FAM significantly increased the protein expression of GPR43, GPR41 and IL-22 (Figure 6B). The hydrocarbon receptor (AhR) and hypoxia-inducible factor 1α (HIF-1α) are considered as the essential factors involved in SCFAs-GPR41-IL-22 pathway in CD4+ T cells. We therefore determined the relative protein expression of intestinal mucosal AhR and HIF-1α. Compared to the control and antibiotic group, the FAM group had higher protein expressions of AhR and HIF-1α (Figure 6B), indicating FAM might upregulate of intestinal microbiota-derived SCFA and modulate IL-22 production in CD4+ T cell in two pathways, SCFAs-GPR41-AhR/HIF1α-IL-22 pathway and SCFAs-GPR43-IL-22 pathway. Moreover, the protein levels of GPR43, GPR41, AhR, HIF-1α and IL-22 in the antibiotic group were significantly lower than those in the control and FAM group.

## 4. Discussion

Weaning stress induces intestinal barrier dysfunction and immune dysregulation in mammals. Here, we use a piglet model to explore the effects of FAM (a probiotic cocultured by *Lactobacillus acidophilus* and *Bacillus subtilis*) on intestinal mucosal barrier from the perspective of metabolic function of gut microbiota. Lactic acid bacteria and bacillus spores have been used as probiotics for human and animal consumption due to their immunostimulatory properties on intestinal immune system [29]. Using single *Lactobacillus acidophilus*, *CRL 1014* was reported to influence lactobacilli community composition and microbial metabolism, thereby improving gut health in humans [30]. Single *Bacillus subtilis* reduced the Escherichia coli infection and diarrhea incidence in weaned pigs via modulating intestinal microbiota, immune response and blood metabolomics [31]. However, combination of bacterial strains that complement each other and occupy different niches within the gut microbiota environment could lead to an enhancement of the effects on the host immune response and health [32]. Coculture of *Lactobacillus* with *Bacillus cereus* could stimulate the biosynthetic capacities of lactic acid strains [33]. Combination of *Bacillus coagulans BC198* and *Lactobacillus paracasei S38* appeared to show greater efficacy in reducing body fat accumulation and modulating gut microbiota than single strains even at a relatively low dose [34]. Another study reported that the effects of oral administration of *Lactobacillus delbrüeckii* and *Bacillus subtilis* combined is better than single on gilthead seabream cellular innate immune responses [32]. Our results showed that co-cultures of *Lactobacillus acidophilus* and *Bacillus subtilis* (FAM) increased body weight and decreased diarrhea incidence in piglets, indicating FAM exerted protective roles in weaning stress.

Intestinal barrier functions as a selective construction to prevent environmental antigens invasion, which is critical for immune resistance and host survival [1]. The development of intestinal barrier occurs rapidly after birth and is characterized by decreased gut permeability [2]. Weaning stress induces intestinal barrier dysfunctions, including defects of the intestinal epithelial junction, decreased thickness of the mucosal layer, and defective production of antimicrobial peptides [8]. In the present study, villi injury and impaired epithelial junctions were observed in weaned piglets. FAM treatment received an improvement in intestinal morphological structure and expression of tight junction protein (Occludin and ZO-1) and AMPs. Occludin and ZO-1 are the main transmembrane proteins in the apical junctional complex (AJC), which regulate the paracellular diffusion of ions and small molecules across epithelial barriers [35,36]. It has been reported that disorder of the AJC contributes to the impaired epithelial barrier function and is a common feature of numerous inflammatory diseases [37]. The increased expression of Occludin and ZO-1 in the present study indicated that epithelial permeability barrier was improved following FAM treatment. Goblet cells and their secretion products mucus are essential portions in mucosal barrier to defense environmental antigens [38]. It has been reported that a Bacillus licheniformis—*B. subtilis* mixture could increase the number of goblet cells in the ileum. Our study showed that FAM increased goblet cells and MUC2 levels, evidencing an enhanced mucosal barrier in FAM piglets. PBD-2, PBD-3 and RegⅢ γ belong to a family of antimicrobial peptides, which stimulate innate immunity [39]. PBD-2 could alleviate inflammation through interacting with Toll-like receptor 4 and suppressing the downstream NF-κB signaling pathway [7]. RegIIIγ has canonical C-type lectin domains that bind to the peptidoglycan of the bacterial cell wall and has direct antimicrobial activity against Gram-positive bacteria, thus protecting epithelial barrier function of intestinal mucosa [28]. PBD-2, PBD-3 and RegⅢ γ levels in the FAM group were higher than the control group, indicating the protective effects of FAM on mucosal barrier. Moreover, FAM reduced the serum level of endotoxin, DAO and D-lactate, indicating that FAM could promote the development of mucosal barrier function and decrease intestinal permeability. 

Studies have demonstrated that probiotics can regulate the structure and diversity of gut microbiota in the host [40]. Our present study showed that observed species, Chao1, ACE and PD whole tree of gut microbiota in FAM-treated piglets were significantly increased, suggesting FAM could improve the α-diversity of gut microbiota. Regarding structural alterations of the microbial community, a study showed that probiotics could increase the Firmicutes: Bacteroidetes ratio and reduce diarrhea in a piglet model [41]. In the present experiment, the Firmicutes was increased while the Bacteroidetes was decreased following FAM treatment, consistent with the previous study. In addition, LEfSe analysis further revealed FAM markedly increased the relative abundance of 18 bacterial biomarkers including Clostridiales, Ruminococcaceae, Firmicutes and Muribaculaceae and decreased relative abundance of 22 microbial biomarkers involving swine Streptococcus, Bacteroides, and Megamonas. Ruminococcaceae is the main bacteria that convert primary bile acids into secondary bile acids and further facilitated intestinal absorption and innate defense [42,43]. The Genomic analysis revealed that Muribaculaceae (S24-7) had functional diversity in degrading complex carbohydrates [44]. The results suggested that FAM modulated microbial diversity and promoted the relative abundance of beneficial microbiota. 

To explore the connection between FAM-induced microbial alterations and modulation of intestinal barrier function, functional metagenomics prediction of gut microbiota was conducted based on 16S rRNA gene sequencing. We found FAM altered 18 pathways at the third level of KEGG pathway including ‘butanoate metabolism’, ‘pyruvate metabolism’ and ‘propanoate metabolism’. Spearman’s correlation analysis further showed higher proportions of bacteria (phyla Firmicutes and genera Clostridium) related to ‘butanoate metabolism’, ‘pyruvate metabolism’, ‘propanoate metabolism’ and ‘fatty acid metabolism’, indicating that FAM might have rebuilt intestinal barrier homeostasis through modulating the metabolic function of gut microbiota. To further confirm the changes in metabolic function of gut microbiota, we performed multiple MS platform-based untargeted metabolomic analysis. The metabolome of colon contents in FAM group was associated with a marked increase in palmitoleic acid, lithocholic acid and squalene, which showed beneficial effects on host immunity and inflammation [45,46,47]. Furthermore, metabolite set enrichment analysis demonstrated amino acid metabolism, aminoacyl-tRNA biosynthesis and short chain fatty acid (SCFA) metabolism were significantly affected following FAM. SCFAs are major metabolic products of intestinal microbiota from dietary fiber and have been recognized as important mediators in regulating mucosal barrier function [40]. The alteration of SCFA metabolism in untargeted metabolomic analysis was in consistent with the function prediction of the gut microbiota, suggesting FAM regulated the SCFA metabolism of gut microbiota in weaning piglets.

To further confirm the changes of SCFA metabolism detected by untargeted metabolomic analysis, we performed a targeted metabolomic analysis to quantify SCFA metabolism. Previous studies have shown that probiotics exerted protective effects on gut health in humans and animals through modulating SCFAs production by bacteria [48,49,50]. In the present study, the production of butyric acid and total SCFAs in colon content was significantly increased following FAM. SCFAs are regarded as important mediators in the communication between the intestinal microbiome and the immune system [51]. Microbiota-derived SCFAs have been demonstrated to protect against several diseases by regulating different immune cells [52,53,54]. Therefore, we detected mucosal immune functions and inflammatory responses in colon tissues. Intestinal-associated lymphoid tissues and antibodies were the important components of mucosal immunity. The differentiation of B cells into SIgA+ cells needed the cytokines secreted by CD4+ T cells and the SIgA+ cells secreted SIgA to the mucus. In the previous study, formula-fed infants receiving a probiotic supplementation maintained higher fecal SIgA levels at the treatment period [55,56]. In the present study, we observed higher levels of CD4+ T cells and SIgA+ cells in intestinal mucosa and SIgA in colon contents following FAM treatment, which was consistent with previous study. A recent study reported that microbiota- derived SCFAs could regulate CD4+ T cell IL-22 production and gut immunity [54]. In our study, FAM decreased the expression of proinflammatory genes and increased the expression of anti-inflammatory gene Il22, indicating FAM strengthened mucosal immunity by modulating SCFAs metabolism and IL-22 production. SCFAs have been shown to reduce intestinal inflammation through regulating host immunity, in which the SCFAs-GPR pathways is considered as one of the most vital mechanisms. Free fatty acids receptors (FFARs) belong to G-protein coupled receptors (GPCRs) including GPR41, GPR43 and GPR109A, which are mainly expressed in intestinal epithelial cells and immune cells and activated by SCFAs [57]. A recent study found that butyrate could promote IL-22 production in CD4+ T cells and ILCs through GPR41 and HDAC inhibition, where AhR and HIF-1α participated in IL-22 production [58]. In our study, we evaluated the expression of IL-22 and short-chain fatty acid receptors including GPR43, GPR41and GPR109A. The results showed that GPR41 and GPR43 expression as well as AHR and HIF1α expression in FAM-treated piglets had a significant increase, suggesting the SCFAs-GPR41 pathway might participate in IL-22 upregulation. The interaction of mucosal IL-22 and intestinal microbiota was pivotal in keeping intestinal homeostasis. On one hand, IL-22 adjusted the gut microbiota composition to improve intestinal barrier. On the other hand, the metabolites of gut microbiota promoted the IL-22 production [58,59].

## 5. Conclusions

In conclusion, our results demonstrated that FAM modulated intestinal barrier homeostasis and immune function though regulation of gut microbiota and SCFAs. GPR41, GPR43, AhR and HIF1α might mediate in the butyric acid induction of IL-22. Thus, our study provided FAM as a novel probiotic intervention in the regulation of gut homeostasis and may give new insights in helping to understand their potential mechanisms underlying in animals or humans.

## Data Availability

Not applicable.

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
