# Peer review of "Co-Cultures of Lactobacillus acidophilus and Bacillus subtilis Enhance Mucosal Barrier by Modulating Gut Microbiota-Derived Short-Chain Fatty Acids"

_nutrients, 2022, doi:10.3390/nu14214475_

Round 1
Reviewer 1 Report
The study assessed the immune effect of Lactobacillus acidophilus and Bacillus subtilis (FAM) on piglet gastrointestinal system. Overall, the study is well presented and the observations are interesting. However, the following issues need to be further modified and improved to meet the criteria of a highly-cited journal. The specific comments as follows:
-
The introductory part is poorly written. Authors should extensively cite relevant research from international counterparts to find out the progress and shortcomings of current research work in this field. For example, tight junctions and microbiota diversity are important for the gut immune system. Nonetheless, the authors do not address the importance of tight junctions or microbiota diversity in the Introduction, but only in the Discussion.
-
Previous studies have shown that Lactobacillus acidophilus and Bacillus subtilis have modulating effects on the gut microbiota; however, most papers used a single organism in their studies. Why the authors use co-cultured organisms instead of individually? Are there any studies showing that co-culture organisms are better than monoculture?
-
Figure 3A shows the results of 16S rRNA gene sequencing. What do the X and Y axes represent? The authors should explain more about the results in Figure 3A.
-
Line 349 is difficult to understand. Also, obviously "protein levels of GPR43, GPR41, AhR, HIF-1α and IL-22 in the antibiotic group" is an incomplete sentence, please check again.

Author Response
Thank you for reviewing our manuscript. We have revised the manuscript according to the comments and outline these changes with red in the re-submitted manuscript. Thanks for your suggestions.
- Thank you. we have added in the discussion.
- Thank you. we have explained in the discussion.
- Thank you. we have revised.
- Thank you. we have checked and corrected.

Reviewer 2 Report
The manuscript “Co-cultures of Lactobacillus acidophilus and Bacillus subtilis enhance mucosal barrier by modulating gut microbiota-derived SCFAs”. The study is good and the findings are satisfactory. However, a few points need to address before the acceptance.
1. Add the full name in the title for the abbreviation “SCFAs”. Similarly, write the full name of the abbreviation DAO in the abstract (line 21)
2. Delete the word gain after “increased body weight” (line 20).
3. Write 28 d as 28 days (line 60). Similarly, correct the 7-d (line 71), and 30-d (line 71).
4. Mention what is the average initial weight of the piglets. Also, mention the average diet, and the time of the diet given to the piglets during the experimental period. It would be better if the author can provide a schematic digram for the section “2.1. Animals and experimental design”.
5. Provide the full name of the abbreviation while they appear first in the MS for example PDB, & Reg III (line no. 95).
6. FAM-treated piglets had higher expression levels of tight junction proteins (ZO-1, and Occludin) and MUC2 protein. Discuss the role of these proteins in association with gut health.
7. Usually, porcine beta defensin-2 (PBD-2), and porcine beta defensin-3 (PBD-3) are the defensive part of the immune system induced to counter the challenge posed by the pathogen. How an author can justify their finding? Add the statement to the discussion.
8. Write the full name of C, F, and A as the control group, FAM group, and antibiotic group, respectively inside figure1, 5, and 6.
9. Figure 6 is wrongly coated as Figure 5 (line 352). Rectify the typo mistake.
10. Figures 3 and 4 are very blurry, improve the resolution and font size.
Author Response
Thank you for reviewing our manuscript. We have revised the manuscript according to the comments and outline these changes with red in the re-submitted manuscript. Thanks for your suggestions.
- Thanks for your suggestion. We have deleted.
- Thank you. We have corrected.
- Thank you for your suggestions. We have revised.
- Thank you. We have provided.
- Thank you. We have discussed.
- Thank you. We have added related statement.
- Thank you. We have added.
- Thank you very much. We have rectified.
- Thank you. We have improved.

Round 2
Reviewer 2 Report
Asked changes have been added I recommend for the publication of the article.